# Association of Stunting with Socio-Demographic Factors and Feeding Practices among Children under Two Years in Informal Settlements in Gauteng, South Africa

**DOI:** 10.3390/children10081280

**Published:** 2023-07-25

**Authors:** Zandile Kubeka, Perpetua Modjadji

**Affiliations:** 1Department of Public Health, School of Health Care Sciences, Sefako Makgatho Health Sciences University, 1 Molotlegi Street, Ga-Rankuwa, Pretoria 0208, South Africa; 2Non-Communicable Diseases Research Unit, South African Medical Research Council, Tygerberg, Cape Town 7505, South Africa

**Keywords:** stunting, socio-demographic factors, feeding practices, children, informal settlements, South Africa

## Abstract

Despite improvements in childhood undernutrition through integrated nutritional programs in South Africa, stunting among children remains persistent, and is attributed to poor socio-demographic status. This context has been implicated in disrupting mothers’ decisions regarding effective infant feeding, ultimately meaning that children’s nutritional demands remain unmet. In view of this, we conducted a cross-sectional study to determine the association between socio-demographic factors and infant and young child feeding (IYCF) practices and stunting among children under two years receiving primary health care in informal settlements in Gauteng, South Africa. A validated questionnaire was used to assess mothers’ socio-demographic status and feeding practices using WHO core indicators. Stunting was defined as length-for age z-scores (LAZ) below −2 standard deviation, computed using WHO Anthro software version 3.2.2.1 using age, sex, and anthropometric measurements of children. Univariate and multivariate analyses were stratified by stunting to determine the relationship with socio-demographic, infant, and IYCF factors using STATA 17. The prevalence of stunting was 16% among surveyed children under two years (with a mean age of 8 ± 5 months) living in poor socio-demographic households. Poor feeding practices were characterized by delayed initiation of breastfeeding (58%), sub-optimal exclusive breastfeeding (29%), discontinued breastfeeding (44%), early introduction of solid foods (41%), and low dietary diversity (97%). Significant differences in terms of child’s age, monthly household income, and ever being breastfed were observed (Chi square test and univariate analysis). After controlling for potential confounders, stunting was significantly associated with child’s age [12–23 months: AOR = 0.35, 95% CI: 0.16–0.76], and monthly household income [ZAR 3000–ZAR 5000: AOR = 0.47, 95% CI: 0.26–0.86]. Despite the few aforementioned socio-demographic and IYCF factors associated with stunting, this study reiterates stunting as the commonest poor nutritional status indicator among children under two years, suggesting the presence of chronic undernutrition in these poverty-stricken informal settlements. A multisectoral approach to address stunting should be context-specific and incorporate tailor-made interventions to promote optimal infant-feeding practices. Conducting future nutrient assessments focusing on children is imperative.

## 1. Introduction

Infant and young child feeding (IYCF) is a set of practices recommended by the World Health Organization (WHO) to promote optimal child growth and development [1]. These feeding practices entail the initiation of breastfeeding within one hour of birth, exclusive breastfeeding for the first six months, and the introduction of nutritionally adequate and safe complementary foods at six months, together with a continuation of breastfeeding for up to two years of age or beyond [2]. Considering their importance, appropriate IYCF practices have been reported to prevent almost 19% of all deaths of under-fives [3]; however, inappropriate IYCF remains a concern in developing countries [2]. It is worth noting that these feeding practices are known to be affected by diverse factors such as family-related circumstances [4], parental knowledge, beliefs, and practices [5], food preparation and preservation, hygiene practices [6], and HIV infection, which can conflict with mothers’ choices regarding breastfeeding [7].

In South Africa, mothers’ feeding practices are sub-optimal, which form one of the major public health problems, despite the country having excellent policies in place and a political commitment to improve infant health and nutrition [8,9]. Studies have reported early breastfeeding initiated within one hour of birth (67.3%), a deficit in exclusive breastfeeding (31.6%), and the introduction of early complementary foods (85%) and mixed feeding (90%) [10,11]. Inappropriate feeding practices, especially among infants from six months of age transitioning from a milk diet to complementary foods, predispose them to unfavorable health outcomes and poor physical growth and development [12,13,14], which continue to exist and vary by sex and setting [12,13,15,16,17].

In particular, stunting (i.e., linear growth retardation or low length-for-age) remains the most common manifestation of undernutrition affecting children under five years amidst concerns regarding inappropriate feeding practices [17]. Local studies have reported stunting as the most persistent indicator for poor nutritional status among children, estimated at a prevalence between 45.3% and 55% [12,16]. Occasionally, stunting among children is accompanied by underweight (i.e., low weight-for-age) and thinness (i.e., low body mass index-for-height) [17], as well as overweight/obesity (i.e., overnutrition) [12,13,15,18,19,20]. The complexity of stunting characteristics can be explained through the multi-level factors contributing to stunting, which include poor socioeconomic status, poor maternal health, and nutrition, inadequate IYCF, and micronutrient deficiencies [21]. Stunted children experience a reduced lean body mass and diminished intellectual functioning, a short adult stature and reduced earnings later in life [22,23], and an increased risk of becoming overweight/obese in adulthood and developing non-communicable diseases (NCDs), which might be exacerbated by weight gain after the age of two years [24,25,26]. Therefore, due to various influences, a deficit in growth status in early life predisposes children to a slow linear growth during middle childhood [27].

Most concerning is the mushrooming of informal settlements in South Africa due to rapid urbanization, which makes this population susceptible to poverty and resultant ill health [28,29,30,31]. Informal settlements come with a poor infrastructure characterized by poor sanitation and electricity issues that make food storage difficult, as well as limited health services and other services, posing special risks to children [32]. Despite the spread of informal settlements, several public health issues, such as feeding practices and poor nutritional status, are less studied in these areas. However, we acknowledge the research conducted on cardiometabolic risks and quality of life [30,31,33,34,35]. This study determined the association between socio-demographic factors and IYCF practices and stunting among children under two years attending primary health care (PHC) facilities in informal settlements in Gauteng, South Africa. This study highlights the gap in IYCF practices in relation to the recommendations of the WHO among a group of children living in informal settlements. The importance of promoting optimal feeding practices to improve nutritional status of children is self-evident, as well as the need for efforts to strengthen contextual programs that educate mothers on feeding practices.

## 2. Materials and Methods

### 2.1. Study Design and Conceptual Frameworks

A cross-sectional study was conducted to determine the association of socio-demographic factors and IYCF practices with stunting among children under two years attending PHC facilities in the informal settlements in Gauteng, South Africa. The study adopted the core indicators for assessing optimal IYCF [36], entailing early initiation of breastfeeding, exclusive breastfeeding under six months, continues breastfeeding at 1 year, introduction of solid, semi solids, or soft foods, dietary diversity, minimum meal frequency, minimum acceptable diet, and consumption of iron-rich or iron-fortified foods. Additionally, the WHO framework for stunting was integrated explaining the factors predisposing to stunting from household and family, feeding practices, and infection [37]. The study was conducted between 2017 and 2018.

### 2.2. Study Setting and Population

This study was conducted at Ekurhuleni District situated in Gauteng Province, South Africa. Ekurhuleni is one of the five districts of Gauteng Province, and one of the eight metropolitan municipalities of South Africa with a total population of 3,500,231 based on the Mid-Year Population Estimates 2016 Statistics South Africa [38,39]. Ekurhuleni has about 91 PHCs, and 11 of them are in Ekurhuleni North 1 sub-district [38]. The study population included infants and their mothers attending PHCs for immunization and growth monitoring in Ekurhuleni North 1 clinics situated in the informal settlements. Gauteng province is one of three out of the nine provinces in South Africa with the highest number of households living in informal settlements [40]. Informal settlements are defined as unplanned residential areas where housing, shelter, and services have been constructed on land that the occupants occupy illegally, according to the United Nations [41]. These unplanned residential areas in South Africa are accompanied by poor health and socioeconomic status [29].

The study included children aged under two years who were brought to the PHC facilities by their biological mothers. Children diagnosed with medical conditions that would have interfered with their feeding and growth status and brought to PHC facilities by non-biological mothers were excluded from the study, as well as mothers who were not able to exclusively breastfeed due to health issues, not willing to participate in the study, and below the age of 18 years.

### 2.3. Sample Size and Sampling Procedure

Using the Cochran formula in a validated sample size calculator (Raosoft Inc., Seattle, WA, USA), a minimum recommended sample for this study was 370, and increased to 444 through a 20% buffering, to cater for a non-response. The total population of children attending PHC facilities in Ekurhuleni North 1, according to DHIS 2015/2016, was estimated to be about 9951 in all 11 clinics in Ekurhuleni north 1 sub-district. We considered an estimated population of children receiving care at PHC facilities, 95% confidence level, and a margin of error of 5% to estimate a representative sample of children in this sub-district [38]. In all, 450 mothers were recruited while waiting in queues for health services in the PHC facilities. Mothers who agreed to participate in the study were requested to come to the arranged space within the health facility after consultation for further engagements. The purpose of the study was explained to the selected mothers, prior to data collection. Out of the recruited 450 mothers, 448 responded, making a 99% response rate. At data capturing, we discovered that 15 of the participants were caregivers, and not biological mothers; meanwhile, at data analysis, six questionnaires had missing data above 10%, as a result, twenty-one questionnaires were excluded. The final sample size of the study was 427 mother–child pairs. First, the three clinics were purposively selected out of the eleven clinics in Ekurhuleni North 1 sub-district, based on the highest population headcount attending PHCs for immunization. Second, within the selected clinics, mothers were selected based on their availability. Hence, convenience sampling, applicable in clinical research [42], was used to select mothers and their infants, following the difficulty in obtaining a random sampling due to long queues in the health facilities, impatience, and resistance. Resistance, low participation of informal settlers in research, and community have been documented due to poor service delivery [43]. A flow chart for recruitment and sampling processes is presented in Figure 1.

### 2.4. Data Collection and Tools

#### 2.4.1. Feeding Practices

Data on infant feeding practices were collected using the WHO questionnaire, estimated from self-reported 24-h recall [44]. The eight core indicators for assessing optimal IYCF, which informed the data collected on infant feeding practices in the current study are [44]:“Early initiation of breastfeeding—a proportion of children born in the last 24 months who were put to the breast within one hour of birth) based on historic recall.Exclusive breastfeeding under 6 months—a proportion of infants 0–5 months who are fed exclusively with breastmilk, based on a 24-h recall/recall of the previous day.Continues breastfeeding at 1 year—a proportion of children 12–15 months who are fed breastmilk based on a recall of the previous day.Introduction of solid, semi solids or soft foods—a proportion of infants 6–8 months who received solid, semi solids or soft foods based on a recall of the previous day.Dietary diversity—a proportion of children 6–23 months who received foods from 4 or more food groups based on a recall of the previous day.Minimum meal frequency—minimum of 2 meals for infants 6–8 months and 3 meals for infants 9–12 months if breastfed; minimum of 4 meals aged 6–12 months if not breastfed.Minimum acceptable diet—infant ≥ 6 months meeting minimum requirements for dietary diversity and meal frequency.Consumption of iron-rich or iron-fortified foods—proportion of children 6–23 months of age who receive an iron-rich food or iron-fortified food that is specially designed for infants and young children, or that is fortified in the home during the previous day”.

#### 2.4.2. Socio-Demographic Variables

Socio-demographic data were collected using a structured questionnaire adapted from the literature [12,16]. Information on children’s date of birth, gender, and illness history were collected in addition to mothers’ household demographics and socioeconomic status through interviews conducted by the main researcher and trained research assistants. The questionnaire was validated through face and content validity, translation, and a pilot study [45]. The questionnaire was first prepared in English and then translated into isiZulu and Sesotho (i.e., a mixture of Sepedi and Setswana) to make it easy for the mothers to understand the questions. The question was further validated through content and face validity, and a pilot study [46]. Independent translators, who spoke isiZulu and Sesotho as their mother tongue and were conversant with English, completed the forward and backward translations of the questionnaire. An expert committee on public health nutrition studies approved the final version of the translated questionnaire. A pilot study was conducted to pre-test the questionnaire and determine its feasibility, among 30 mothers who did not participate in the main study, estimated to be a suitable number for piloting a concept. In addition to training the research assistants to administer the questionnaire and measure anthropometry, they were moderated while conducting preliminary interviews in local languages during piloting. Following a pilot study, only minimal changes to the study were implemented in terms of layout and style, and the results were not included in the main study.

#### 2.4.3. Anthropometric Measurements and Nutritional Status Indicators of Children

Trained research assistants collected children’s weights and heights according to WHO procedures [47]. Weight (W) was measured to the nearest 0.1 g using a Seca 354 baby electronic scale by Medicare hospital equipment in South Africa, manufactured in Germany. This measurement entailed the child’s length lying down (recumbent) on the scale placed on a flat, stable surface such as a table. The recumbent length of each child was measured in a supine position to the nearest 0.1 cm using the Seca 210 measuring mat. Measurements were recorded on an anthropometry sheet and captured on the WHO Anthro software version 3.2.2.1 to computed Z-scores based on the 2006 WHO growth standards, expressed as standard deviation units from the median value for the WHO growth reference groups taking age and sex into consideration. From the length measured above from children, stunting was computed as length-for-age z-score (LAZ) below −2 standard deviation (SD). From the weight measured above from children, underweight was computed as weight-for-age z-scores (WAZ) below −2SD, and from the generated body mass index (BMI) of children, thinness was computed as BMI-for-age z-scored (BAZ) below −2SD, and overweight and obesity as BAZ > 2SD and ≤ 3SD, and BAZ > 3SD, respectively [47,48].

### 2.5. Data Analysis

STATA version 17 (Stata Corp. 2015. Stata Statistical Software: Release 17, College Station, TX, USA) was used to compute descriptive and inferential statistics. Missing data were detected through complete case analysis. Tabulations of household and maternal socio-demographic variables were performed through frequencies (*n*) and percentages. Distributions of continuous variables, such as age, anthropometry, and z-scores were checked using the Shapiro–Francia test, and results presented as median (interquartile range (IQR)]. Data analysis was stratified by stunting as the outcome variable; meanwhile, the numerical data were categorized for comparison purposes, easier interpretation, results, and the display of descriptive statistics. Non-parametric tests, Mann–Whitney U and Kruskal–Wallis, were used to compare the medians between two groups and three groups by sex and age groups, respectively. A Chi-square test (and Fischer’s exact test for cells with less than five cases) were used to compare non-stunted and stunted children by mothers’ and children’s characteristics. Univariate [crude odds ratio (COR); *p*-value ≤ 0.25] and multivariate analyses [adjusted odd ratio (AOR); *p*-value < 0.05] were used to determine the associations of stunting and selected covariate.

## 3. Results

### 3.1. The Nutritional Status of Children

The study consisted of 427 mother–child pairs. Out of 427 children, 210 (49%) were boys and 217 (51%) were girls. Children were further divided into three age groups; younger (0–5 months; *n* = 176 (41%)), middle aged (6–11 months; *n* = 145 (34%)), and older (12–23 months; *n* = 106 (25%)) children with a mean age of 8 ± 5 months. In Figure 2, nutritional status indicators (i.e., stunting, underweight, and thinness) among children are plotted by sex and age groups. No significant differences were observed for median LAZ, WAZ, and BAZ by sex (using the Mann–Whitney test) and age (using the Kruskal–Wallis test).

### 3.2. Prevalence of Stunting, Underweight and Thinness

Figure 3 describes and compares the prevalence of stunting, underweight, and thinness of children by age and sex groups using Chi-square test and Fischer’s exact for cells with less than five cases. LAZ yielded the prevalence of children with normal z-scores, stunted, and tall, while WAZ yielded the prevalence of children with normal z-scores, underweight, and growth problems, and BAZ yielded the prevalence of children with normal z-scores, thinness, overweight risk, overweight, and obesity. During analysis, we considered normal children, and those with stunting, underweight, and thinness. In all, 16% of children were stunted, 10% were underweight, and 3% were thin. Prevalence of stunting (*p* = 0.017) and underweight (*p* = 0.020) were significantly different by age and by sex.

### 3.3. Stunted/Non-Stunted Children and Characteristics of Mothers

Mothers were divided into two age groups; younger mothers aged below 30 years (*n* = 260 (61%)) and older mothers aged 30 years and above (*n* = 167 (39%)), with mean age of 29 ± 6 years ranging between 18 and 45 years. Most mothers were single (59%), had secondary education (70%), were unemployed (77%), and living in larger households (38%). Household income was categorized into three groups, which were low (<ZAR 3000), middle (ZAR 3000–ZAR 5000), and high (>ZAR 5000) income, and the low (40%) and middle (47%) income were common. We roughly considered that the median wage in South Africa is ≈ZAR 3000 per month, which is considered low [49]. LAZ (*n* = 427) yielded the prevalence of children with normal z-scores, stunted, and tall, and during analysis, we considered normal children (*n* = 345) and stunted (*n* = 65), and excluded tall children (*n* = 17). Stunted and non-stunted children were compared by the demographics of their mothers using Chi-square test and Fischer’s exact test for cells with less than five cases. Significant difference of stunted children was observed by monthly household income (*p* = 0.024) and over half of stunted children (51%) lived in households with a low income compared to non-stunted children (39%) (Table 1).

### 3.4. Stunted/Non-Stunted Children and Their Characteristics

LAZ (*n* = 427) yielded the prevalence of children with normal z-scores, stunted, and tall, and during analysis using Chi-square test and Fischer’s exact test for cells with less than five cases, we considered normal children (*n* = 345) and stunted (*n* = 65), and excluded tall children (*n* = 17). Most children (*n* = 415 (98%)) were born in health facilities, and only few (*n* = 47 (11%)) were attending day-care. Significant differences of stunted children were observed for child’s age (*p* = 0.017) and being ever breastfed (*p* = 0.038) compared to the non-stunted children. Most stunted children were aged below 5 months (55%) and between 6 and 11 months (39%), while few stunted children (8%) were ever breastfed compared to the non-stunted (8%) (Table 2). Further results showed no significant differences were observed for IYCF practices between stunted and non-stunted children, but results showed that more of the non-stunted children compared to stunted children were initiated to breastfeeding within one hour of birth in this study. Feeding practices of children who were characterized by initiation of breastfeeding within one hour was observed in one-third of children (31%), exclusive breastfeeding (29%), early introduction of solid foods (41%), and low dietary diversity in almost all children.

### 3.5. Association of Stunting with Covariates

Table 3 shows the association of stunting and selected socio-demographic factors and IYCF practices variables. Univariate logistic regression (COR) showed associations of stunting with child’s age, household income, initiation of breastfeeding, and being ever breastfed (*p* ≤ 0.25). After controlling for potential confounders (socio-demographic, anthropometric, and feeding practice variables), the final hierarchical logistic regression showed significant associations of stunting with child’ age [12–23 months: AOR = 0.35, 95% CI: 0.16–0.76], and monthly household income [middle-income: AOR = 0.47, 95% CI: 0.26–0.86]. No significant associations were observed between stunting and feeding practice variables in the multivariable model.

## 4. Discussion

Investigating stunting among children who may be at risk of compromised physical growth and development can never be over-emphasized as one of the public health concerns in South Africa due to its early and long-term consequences and implications. The first 1000 days of life from conception to two years of age are risky for growth and development among children because of the associative link with feeding practices that may either have positive or negative long-lasting consequences [50]. Therefore, in this study, we determined the association of socio-demographic factors and IYCF practices with stunting among children under two years attending PHC facilities in informal settlements in Gauteng, South Africa.

Prevalence of stunting (16%) among children was high in the district, despite nutritional programs put in place to improve the nutritional status of South African population groups [13,15,19,20]. This prevalence of stunting was either lower or higher in comparison with the prevalence reported in other local studies by settings and regions [12,13,16,18,20]. For instance, different levels from low, moderate to high prevalence of stunting among children have been recorded in Limpopo (3.7–45.3%) [12,18], Mpumalanga (51%) [20], North-West (29%) [13] and Gauteng (55%) [16] provinces. The same public health challenge of persistent moderate to high prevalence of stunting has been observed in African countries like Ethiopia (37.9%) [51], and Kenya (24%) [52]. The discrepancies in persistent stunting in this study can be attributed to sample size and/or setting (i.e., context), which obviously might have affected the outcome variable. The consequences of stunting such as poor cognitive and physical development and increased risk of chronic diseases in adulthood [53], make stunting one major issue of concern in South Africa that requires continuous attention, especially during the first two years of life [54].

Regarding feeding practices in this study, delayed initiation of breastfeeding (58%), sub-optimal exclusive breastfeeding (29%), and discontinuation of breastfeeding (44%) are consistent with local studies [14,55,56]. The breastfeeding initiation rate in this study is lower compared to rates reported in South Africa on a national level (88%) [57], in low-income areas (77%) [55], rural and peri-urban areas (88–100%) [14,58,59] and developing countries (95%) [60]. The importance of early breastfeeding initiation immediately after giving birth, establishing milk production, and increasing breastfeeding success has been acknowledged in South Africa [55,59]. It has been well documented that South Africa is behind on the global target of 50% exclusive breastfeeding, currently estimated at 32% [11]. Additionally, early introduction of solid foods (41%), and low dietary diversity (97%) in this study have been reported in other studies [12,13].

The differences between stunted and non-stunted children have been reported regarding growth and development [61], child’s characteristics [62,63], as well as socio-demographic factors [37] and feeding practices [64]. In this study, we found associations of stunting with child’s age and monthly household income, initiation of breastfeeding, and being ever breastfed (chi-square and univariate regression analysis tests), and child’s age and monthly household income (multivariable regression analysis), similar to other reports [11,12,65,66,67,68]. For instance, children below 6 months of age (21%) were more stunted, followed by those aged between 6 and 11 (14%), and 12 and 23 months (9%). Consistent results have been reported in South Africa showing that stunting affected almost a third (28.5%) of 6-month-old infants, [56] and ranges from 11% to 16% among 6–12-month-old infants [69,70], with a similar trend in other African countries [71]. Previous studies in low-and-middle income countries have described that low height in stunted children is a main factor in growth and development delays [72,73]. Like other research linking stunting to breastfeeding [64], we also found that fewer children in the stunting group were ever breastfed compared with the not stunted group. Furthermore, more of non-stunted children compared to stunted children were initiated to breastfeeding within one hour of birth in this study. Literature documents the importance of breastfeeding to supply newborns with colostrum, a deep lemon-colored liquid secreted by the breasts for the first several postpartum days, and compared with mature milk, it (i.e., colostrum) contains more minerals and protein [74], which are necessary for forming the gut microbiota and immune system [61].

Lastly, on the contrary, inconsistencies have been reported regarding the associations of stunting with socio-demographic status, whereby some studies have reported no significant differences between the non-stunted and stunted children [75] while others have indicated clear significances [61,76,77]. But socio-economic factors, mostly including income, have consistently been implicated in the prevalence of stunting in developing countries [78]. We also found an association between household income and stunting in this study. This is not out of the ordinary, considering that stunting develops over a relatively long period and is attributable to household poverty [51]. Poverty in this study is explained by poor living conditions observed through poor socio-economic households’ context, larger numbers of household members, and predominant low and middle household income. Poverty in South African households has been reported [17,79], especially in settings with poor infrastructure like rural and informal settlements [27,31]. Worth noting, is that South Africa has experienced urbanization, which gave rise to informal settlements, poverty, and resultant ill health [29,31]. To date, the country continues to experience poverty and poor socio-economic status, predisposing households to food insecurity through limited availability, access, and affordability [66,80]. Therefore, compromised household food security promotes low dietary diversity, also observed in this study, depriving children of the ability to meet their nutrient requirements for growth, in addition to suboptimal feeding practices implicated in the minimal research reported previously in informal settlements in South Africa [81]. Further reports in other African countries have associated stunting with low wealth index [82] and pointed out that involvement from a spouse in child caregiving and providing income could contribute to the better health of the children [83,84]. Therefore, the high rate of singlehood in this study substantiates lack of spousal support with harmful consequence towards household upkeeping regarding income and food security.

### Limitations

Although the cross-sectional design used in this study rules out causality, inferences are well founded considering the reliability and validity maintained during the research process. Second, we used convenience sampling due the difficulty of obtaining a random sample, and this was mitigated by obtaining a large sample size for the study. Third, some possibly unmeasured confounders and residual confounding coupled with potential reporting and recalling bias and social desirability may have influenced the effect of feeding practices on stunting. Fourth, the limited associations between stunting and covariates may not necessarily be generalizable to every resource-constrained study but rather may be context-specific. We acknowledge not testing the reliability of the questionnaire statistically as one of the characteristics of a valid questionnaire; however, the questionnaire was adapted from credible studies and piloted before use to ensure its reliability. Finally, we did not measure the HIV status of mothers, which could have provided a better indication regarding breastfeeding, in addition to omitting mother’s anthropometric measurements. Nonetheless, our study has been able to determine the prevalence of stunting among children aged under two years in the PHC facilities in selected informal settlements in Gauteng, South Africa, and further associated stunting with the socio-demographic factors and IYCF practices.

## 5. Conclusions

This study reports a high prevalence of stunting among children under two years in selected informal settlements in Ekurhuleni District situated in Gauteng Province of South Africa. Stunting was associated with child’s age, monthly household income, initiation of breastfeeding, and being ever breastfed. Further logistic regression analysis showed that children aged 12–23 months have lower odds of increased risk of stunting compared to children aged between 0 and 5 months. Children living in households with a middle income monthly are less likely to be stunted compared to children living in households with a low income. Despite the few aforementioned socio-demographic and IYCF factors associated with stunting, this study reiterates stunting as the most common poor nutritional status indicator among children under two years, suggesting the presence of chronic undernutrition in these poverty-stricken informal settlements. These findings highlight suspected growth and development delays among these children, who might further be predisposed to lower cognitive function than children who are not stunted because of their stunted brains, which affect their learning abilities and education performance in school as they grow, as well as lack of productivity as they age into adulthood. Therefore, possibilities are that the context of poverty observed in this study disrupts mothers from making decisions for effective infant feeding, ultimately condoning unmet child nutritional demands. The multisectoral approach to address stunting should be context-specific and incorporate tailor-made interventions to promote optimal infant feeding practices among mothers. Future research on nutrient assessments among children is imperative.

## Figures and Tables

**Figure 1 children-10-01280-f001:**
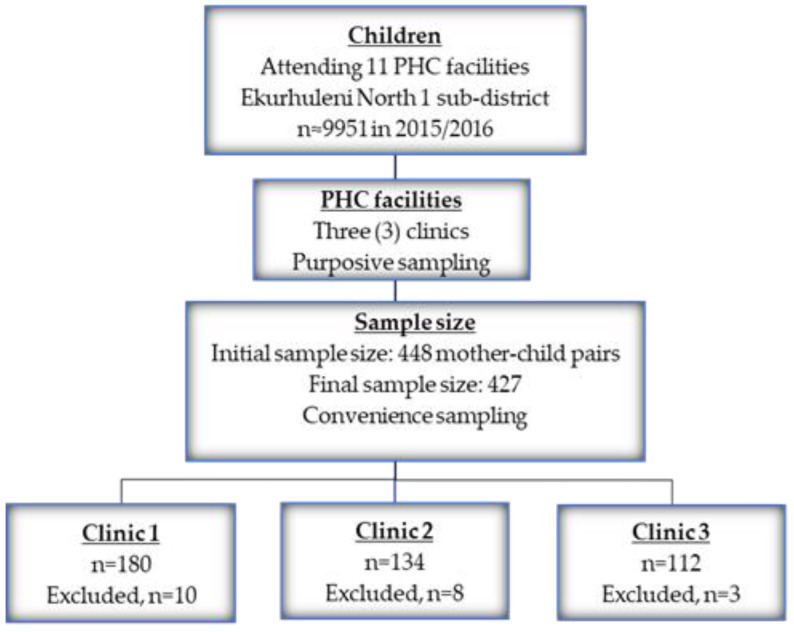
A flow chart on recruitment and sampling processes.

**Figure 2 children-10-01280-f002:**
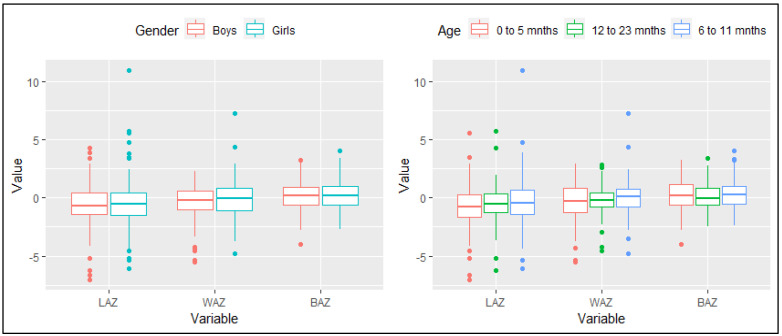
Stunting, underweight, and thinness among children by sex and age groups.

**Figure 3 children-10-01280-f003:**
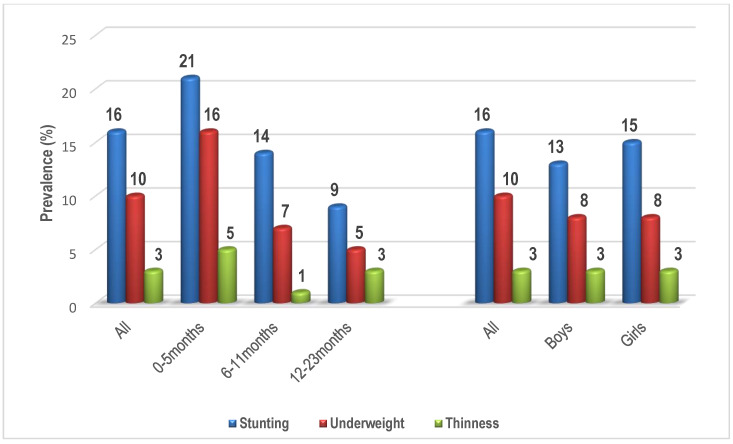
The prevalence of nutritional status indicators of children by age and sex groups.

**Table 1 children-10-01280-t001:** Comparison of stunted/non-stunted children by mothers’ characteristics.

Variables	All (*n* = 427) *n* (%)	Non-Stunted (*n* = 345) *n* (%)	Stunted (*n* = 65) *n* (%)	*p*-Value
Mothers’ age (years)				
<30	260 (61)	211 (61)	42 (65)	0.559
≥30	167 (39)	134 (39)	23 (35)	
Marital status				
Cohabiting	159 (37)	129 (37)	27 (42)	0.806
Single	252 (59)	203 (59)	36 (55)	
Ever married	16 (3)	13 (4)	2 (3)	
Education level				
No education/Primary	41 (10)	30 (9)	7 (11)	0.886
Secondary	301 (70)	245 (71)	45 (69)	
Tertiary	85 (20)	70 (20)	13 (20)	
Employment status				
Unemployed	330 (77)	265 (77)	54 (83)	0.265
Employed	97 (23)	80 (23)	11 (17)	
Household size				
1–4	264 (62)	211 (61)	41 (63)	0.771
≥5	163 (38)	134 (23)	24 (37)	
Household income/month				
Low	170 (40)	134 (39)	33 (51)	0.024 *
Middle	201 (47)	169 (49)	20 (31)	
High	12 (3)	42 (12)	12 (18)	
Type of house				
Brick	6 (1)	4 (1)	2 (3)	0.384
Prefabricated	8 (2)	8 (2)	0	
Corrugated/zinc	114 (27)	93 (27)	16 (25)	
Mud/wood	299 (70)	240 (70)	47 (72)	
Water access				
No	19 (4)	15 (4)	3 (5)	0.923
Yes	421 (96)	330 (96)	62 (95)	
Type of toilet				
Non-flush	85 (19)	66 (19)	12 (18)	0.900
Flush	342 (80)	279 (81)	53 (82)	

* Stands for significant difference, *n* stands for number of participants, and % stands for percentage.

**Table 2 children-10-01280-t002:** Comparison of stunted/non-stunted children by their characteristics.

Variables	All (*n* = 427) *n* (%)	Non-Stunted (*n* = 345) *n* (%)	Stunted (*n* = 65) *n* (%)	*p*-Value
Child’s sex				
Boys	210 (49)	174 (50)	30 (46)	0.527
Girls	217 (51)	171 (50)	35 (54)	
Child’s age (months)				
0–5	176 (41)	132 (38)	36 (55)	0.017 *
6–11	145 (29)	118 (34)	20 (31)	
12–23	106 (25)	95 (28)	9 (14)	
Child’s place of birth				
Health facility	415 (98)	334 (97)	65 (100)	0.144
Non-health facility	12 (2)	11 (3)	0	
Child attending day-care				
No	380 (89)	306 (89)	60 (92)	0.388
Yes	47 (11)	39 (11)	5 (20	
Ever breastfed				
No	32 (7)	28 (8)	1 (2)	0.038 *
Yes	395 (93)	317 (92)	64 (98)	

* Stands for significant level at <0.05, *p* stands for probability level, *n* = number of participants and % stands for percentage.

**Table 3 children-10-01280-t003:** Association of stunting and covariates.

Stunting	COR (95% CI)	*p*-Value	AOR (95% CI)	*p*-Value
Age (months)				
0–5	1		1	
6–11	0.62 (0.34–1.13)	0.120	0.62 (0.34–1.13)	0.120
12–23	0.35 (0.16–0.76)	0.008 *	0.35 (0.16–0.76)	0.008 *
Household income (monthly)				
<ZAR 3000	1		1	
ZAR 3000–R5000	0.48 (0.26–0.88)	0.017 *	0.47 (0.26–0.86)	0.014 *
>ZAR 5000	1.16 (0.55–2.45)	0.696	1.09 (0.51–2.33)	0.821

* Stands for significant level at <0.05, *p* stands for probability level, COR stands for crude odds ratio, AOR stands for adjusted odds ratio, CI stands for confidence interval, and % stands for percentage.

## Data Availability

The dataset for the study group generated and analyzed during the current study is available from the corresponding author upon reasonable request due to ethical restrictions.

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
