# Peer review of "Association of Stunting with Socio-Demographic Factors and Feeding Practices among Children under Two Years in Informal Settlements in Gauteng, South Africa"

_children, 2023, doi:10.3390/children10081280_

Round 1
Reviewer 1 Report
Introduction:
The introduction section is well written. However, the last paragraph is very vague. The authors only examined the socio-demographic and IYCF factors with stunting. Clearly write the objective of the study and potential implications of the study in this paragraph.
Methods:
Measurements of variables: The authors have explained the conceptual definition of variables. If the authors also mention how they measure the variables or categories of the variables for a better understanding of the readers.
Outcome variable: The outcome variable is stunting. However, apart from stunting, the authors also collected data on wasting and underweight. However, the authors did not discuss and correlated with these important variables. Did the authors only interest in stunting? Why the other important nutritional indicators are not studied with other independent variables?
I would suggest that the author can also think to explore the socio-demographic factors with other important nutritional indicators such as child underweight. Review the statistical power of underweight count.
Results:
The authors can present some data through graphs etc., where appropriate. Maybe Table 1 can be presented through some graphs.
Household income: please also clarify the three income groups. Are the authors referring low income, middle-income, and high-income groups keeping in view of mentioned income categories? What is the poverty definition in South Africa?
Discussion:
The author used the word nutritional status in the paper. However, the outcome of this paper is stunting. Maybe the author can use stunting instead of nutritional status.
Author Response
Reviewer 1
The introduction section is well written. However, the last paragraph is very vague. The authors only examined the socio-demographic and IYCF factors with stunting. Clearly write the objective of the study and potential implications of the study in this paragraph.
Response: we have revised the objective of the study; lines 15-17.
Methods:
Measurements of variables: The authors have explained the conceptual definition of variables. If the authors also mention how they measure the variables or categories of the variables for a better understanding of the readers.
Response: Additional explanation has been added in the methods; Lines 175-179, and 181-185
Outcome variable: The outcome variable is stunting. However, apart from stunting, the authors also collected data on wasting and underweight. However, the authors did not discuss and correlated with these important variables. Did the authors only interest in stunting? Why are the other important nutritional indicators not studied with other independent variables? I would suggest that the author can also think to explore the socio-demographic factors with other important nutritional indicators such as child underweight. Review the statistical power of underweight count.
Response: The prevalence of underweight and thinness were very low. We were informed by the results that stunting is the commonest poor nutritional status indicator. Therefore, we saw it valid to focus on stunting in the analysis as an outcome variable considering the low statistical power for underweight and thinness. We rephrased the manuscript to focus on stunting from the title to the conclusion.
Results:
The authors can present some data through graphs etc., where appropriate. Maybe Table 1 can be presented through some graphs.
Response: We have changed tables to box plots; lines 206-207
Household income: please also clarify the three income groups. Are the authors referring low income, middle-income, and high-income groups keeping in view of mentioned income categories? What is the poverty definition in South Africa?
Response: Yes, we have rephrased the three income groups as you advised and added roughly a low wage estimated in South Africa; line 223-225.
Discussion:
The author used the word nutritional status in the paper. However, the outcome of this paper is stunting. Maybe the author can use stunting instead of nutritional status.
Response: We have revised the entire discussion focusing on stunting as the outcome variable; lines 266-329. We amended the title as well.
Reviewer 2 Report
Review report
Comment on key sections
Title: well written, self-explanatory
Abstract: contains key sections of the study.
Suggestion for improvement: make the result section concise (Ln 20 – 27)
Aim of the study: Clearly stated, to determine nutrition status and feeding practices of children living in informal settlements of Gauteng, South Africa (Ln 13 -14)
Introduction: Has a short review of literature, includes definition of feeding terms, and tries to explain why the study was conducted (justification)
Suggestion for improvement: Language editing
Materials and methods
Well written detailed methodology, states study population, Selection criteria, exclusion/inclusion criteria (sample size and procedure) well stated.
Suggestion for improvement: For clarity, summarize the selection criteria in a flow diagram (Ln 123 – 134).
Data collection tools well stated, in detail, pre-tested. Data analysis. detailed data analysis conducted using robust package and recent version (STAT version 17).
Results: well written and presented, the results are reproducible based on the details given in the methods section. The tables and figure are appropriate, properly show the data and easy to interpret and understand. The findings are interpreted appropriately.
Discussion: Findings well discussed and compared with earlier study findings. Study limitation well stated.
Suggestion for improvement-Language editing
Conclusions: The conclusions drawn are coherent and consistent with the evidence and arguments presented, and recommendations
Suggestion for improvement: Include a call for further research.
Reference: Used recent and appropriate references. The cited references recent publications and relevant and does not include an excessive number of self-citations.
.
Author Response
Reviewer 2
Comment on key sections
Title: well written, self-explanatory
Response; noted.
Abstract: contains key sections of the study.
Suggestion for improvement: make the result section concise (Ln 20 – 27)
Response: Results have been revised
Aim of the study: Clearly stated, to determine nutrition status and feeding practices of children living in informal settlements of Gauteng, South Africa (Ln 13 -14)
Response: Noted, revised as per the other reviewer comment
Introduction: Has a short review of literature, includes definition of feeding terms, and tries to explain why the study was conducted (justification)
Suggestion for improvement: Language editing
Response: Taken for English editing
Materials and methods
Well written detailed methodology, states study population, Selection criteria, exclusion/inclusion criteria (sample size and procedure) well stated. Suggestion for improvement: For clarity, summarize the selection criteria in a flow diagram (Ln 123 – 134).
Response: Flow chart added; lines 132-134
Data collection tools well stated, in detail, pre-tested. Data analysis. detailed data analysis conducted using robust package and recent version (STAT version 17).
Response: noted
Results: well written and presented, the results are reproducible based on the details given in the methods section. The tables and figure are appropriate, properly show the data and easy to interpret and understand. The findings are interpreted appropriately.
Response: noted
Discussion: Findings well discussed and compared with earlier study findings. Study limitation well stated. Suggestion for improvement-Language editing
Response: taken for English editing
Conclusions: The conclusions drawn are coherent and consistent with the evidence and arguments presented, and recommendations. Suggestion for improvement: Include a call for further research.
Response: Noted: Call for future research added: line 355-356
Reference: Used recent and appropriate references. The cited references recent publications and relevant and does not include an excessive number of self-citations.
Response: noted.
Reviewer 3 Report
1. Title does not coherent with the study design, results.
2. Continuous variables were changed as categorical variables without any logical reason. Mother's age, household size, household income.
3. When most of the results are not significant, what is novelty in the manuscript? It was not clearly indicated.
4. Conclusion are not directly relevant to the results.
5. Too many reference were cited some without their relevance to the region. Moreover, discussion section explains only the similar results of the reference and the manuscript without other details. That gives the idea that the results have been compared on the basis of 'chance'.
Author Response
Reviewer 3
- Title does not coherent with the study design, results.
Response: Association added in the title.
- Continuous variables were changed as categorical variables without any logical reason. Mother's age, household size, household income.
Response: added in lines 192-194
- When most of the results are not significant, what is novelty in the manuscript? It was not clearly indicated.
Response: added in lines 267 -269, and in lines 352-362
- Conclusion are not directly relevant to the results.
Response: Conclusion revised after revising the title and the discussion.
- Too many references were cited some without their relevance to the region. Moreover, discussion section explains only the similar results of the reference and the manuscript without other details. That gives the idea that the results have been compared on the basis of 'chance'.
Response: We have revised the discussion as per the guidance of the other reviewer, and references have reduced from 100 to 88.
Round 2
Reviewer 3 Report
I appreciate the authors' sincere efforts to improve their manuscript. However, before going for the final decision, I have some more concerns:
1. The manuscript is unnecessarily made lengthy. For example, in Table 3, no results show some evidence of difference. A brief explanation may be added in a paragraph by omitting the table.
2. The plagiarism is very high (29%) with me through turnitin. I would suggest authors to reduce it before resubmission.
3. Little efforts need the attention of authors. Some of these are:
- First two paragraphs of the introduction section reveal almost the same information. Redundant parts may be removed.
- 'The study includes children under two years' has been repeated unnecessarily. It may be removed from some paragraphs.
- overemphasized as one of the (line: 268)
4. Authors stated that they transformed the continuous variables (sex and age) into categories before analyses. At the same time, authors have used the Mann-Whitney U test and the Kruskal Walis test which are meant for continuous data. How authors can confirm the authenticity of the results?
Author Response
Reviewer 3; Round 2
I appreciate the authors' sincere efforts to improve their manuscript. However, before going for the final decision, I have some more concerns:
Response: It is because we had to address the comments of other reviewers as per their advises – with a need for extra information.
- The manuscript is unnecessarily made lengthy. For example, in Table 3, no results show some evidence of difference. A brief explanation may be added in a paragraph by omitting the table.
Response; Table 3 has been deleted and a brief explanation added in the narrative of table 2 – lines 242-246.
- The plagiarism is very high (29%) with me through Turnitin. I would suggest authors to reduce it before resubmission.
Response: The similarity index in this manuscript was 17% (uploaded on unpublishable material) in the previous round – excluding affiliations, authors’ contribution, acknowledgements, ethical statement and funding, conflict of interest and references. We have uploaded another similarity index report (15%) for the third version of the manuscript after revising the paper as per your comments. Note that the Turnitin manuscript excludes the above-mentioned sections, because they can never be amended.
- Little efforts need the attention of authors. Some of these are:
- First two paragraphs of the introduction section reveal almost the same information. Redundant parts may be removed.
Response: revised; lines 36 - 53
- 'The study includes children under two years' has been repeated unnecessarily. It may be removed from some paragraphs.
- overemphasized as one of the (line: 268)
Response: edited - line 264
- Authors stated that they transformed the continuous variables (sex and age) into categories before analyses. At the same time, authors have used the Mann-Whitney U test and the Kruskal Walis test which are meant for continuous data. How can authors confirm the authenticity of the results?
Response: These non-parametric tests; Mann-Whitney is used to compare numerical medians for z-scores; LAZ, WAZ and BAZ between 2 groups (by sex), while Kruskal Wallis compares numerical medians for z-scores; LAZ, WAZ and BAZ between 3 groups (by age) (TABLE 1); line 204-205 and lines 189-190. These we do before we group (categorize) the z-scores (numerical) into references ranges to compute the proportions/prevalence (categorical) of nutritional status indicators.
In Categorical variables, which can either be binary, nominal, or ordinal, including grouping continuous data, proportions are compared using a Chi square test or Fischer’s’ exact test depending on the number of cases per cell - Tables 1 (lines 232- 233) and 2 (lines 248-249). Further use of categorical data is in table 4 where we used logistic regression analysis Table 3 (lines 258-259). Hope this confirms the authenticity of our results. The dataset used for this paper is available upon request. We really appreciated your detailed review. It has surely assisted us to improve this manuscript.